# Prediction of Prognosis in Patients with Trauma by Using Machine Learning

**DOI:** 10.3390/medicina58101379

**Published:** 2022-10-01

**Authors:** Kuo-Chang Lee, Chien-Chin Hsu, Tzu-Chieh Lin, Hsiu-Fen Chiang, Gwo-Jiun Horng, Kuo-Tai Chen

**Affiliations:** 1Emergency Department, Chi-Mei Medical Center, Tainan 710402, Taiwan; 2Department of Biotechnology, Southern Taiwan University of Science and Technology, Tainan 71005, Taiwan; 3Department of Computer Science and Information Engineering, Southern Taiwan University of Science and Technology, Tainan 71005, Taiwan

**Keywords:** trauma, machine learning, prognostic predictor, mortality, trauma score

## Abstract

*Background and Objectives:* We developed a machine learning algorithm to analyze trauma-related data and predict the mortality and chronic care needs of patients with trauma. *Materials and Methods:* We recruited admitted patients with trauma during 2015 and 2016 and collected their clinical data. Then, we subjected this database to different machine learning techniques and chose the one with the highest accuracy by using cross-validation. The primary endpoint was mortality, and the secondary endpoint was requirement for chronic care. *Results:* Data of 5871 patients were collected. We then used the eXtreme Gradient Boosting (xGBT) machine learning model to create two algorithms: a complete model and a short-term model. The complete model exhibited an 86% recall for recovery, 30% for chronic care, 67% for mortality, and 80% for complications; the short-term model fitted for ED displayed an 89% recall for recovery, 25% for chronic care, and 41% for mortality. *Conclusions:* We developed a machine learning algorithm that displayed good recall for the healthy recovery group but unsatisfactory results for those requiring chronic care or having a risk of mortality. The prediction power of this algorithm may be improved by implementing features such as age group classification, severity selection, and score calibration of trauma-related variables.

## 1. Introduction

Trauma is a leading cause of death in adults, especially in the young population, and it remains a considerable public health concern worldwide, including in Taiwan and the United States [1,2,3]. Therefore, many trauma warning systems and predictors have been developed, such as the injury severity score (ISS), revised trauma score (RTS), and trauma and injury severity score (TRISS), which have improved efficiency and efficacy of predicting prognosis of patients with trauma by simplifying evaluation [4,5]. Most studies have focused on the prediction of mortality and resource utility. However, few studies have evaluated the medical requirements for chronic care needs, residual functional performance in various patient subpopulations after trauma, and practical application of predicting calculators for patients with trauma [6,7].

Machine learning (ML) approaches using big data have heralded a new era of improvements in prognosis prediction [8] and quality improvement intervention [7]. In certain medical fields, many ML techniques, such as support vector machines, decision trees, random forests, logistic regression, and AdaBoost, have been developed based on the learning of various complicated and nonlinear interaction variables and have exhibited high, optimized predictive performance of observed outcomes [4,9,10]. In the present study, we developed a practical ML-based model to analyze trauma-related data collected from a level I trauma center and predict the mortality and chronic care needs of patients with trauma.

## 2. Methods

This study was reviewed and approved by the Institutional Review Board of Human Research, Chi-Mei Medical Center (IRB code: 11012-J02, date of approval: 21 June 2022); they waived the requirement for informed consent because the data were anonymized and collected retrospectively. We analyzed a dataset that included all patients with trauma discharged from wards in Chi-Mei Medical Center between 1 January 2015 and 31 December 2016. The dataset included patients’ data regarding demographics (age, sex, and comorbidities), prehospital presentations (locations of injury, mechanisms of injury, deliberate or accidental injury, dispatch of emergency medical service, time and transportation from injury sites to the hospital, prehospital and cardiac arrest, vital signs at the scene, prehospital management by emergency medical technicians, and level of the transferred hospital), emergency department (ED) presentations (vital signs and Glasgow Coma Scale in the ED, triage, serum test of alcohol level, emergency interventions in the ED, activation of trauma team, applications of ultrasound/computed tomography/magnetic resonance imaging, time from the ED to the ward, and time from the ED to the operating room), various trauma scores (abbreviated injury scale [AIS], ISS, new ISS, RTS, and TRISS), hospital course (requirements for surgery, time from the ED to the operation room, time from surgical decision to surgery, requirements for intensive care, length of hospital stay, and length of stay in the intensive care unit), and prognosis (recovery, chronic care needs, mortality, and complications) (Figure 1). Continuous data are presented as means ± standard deviations.

### 2.1. Dataset Management and ML Technique

The data of 80% of the study cohort were considered the training set, and the remaining 20% were the test set by overlaying multiple levels, which is usually called cross-validation, to improve model performance by repeated cycles [10,11]. We preprocess the data before the forecast. Because some values are clustered in certain intervals, they need to go through data cleaning, data conversion, and data reduction. Data cleaning is mainly for correcting loss values. Data conversion mainly works for standardization, attribute selection, and discrete processing. We segmented continuous data into segments for data discretization, which can effectively overcome the hidden defects in the data and make the model results more stable. The main purpose of discrete processing is to map limited individuals of infinite space into limited space. Data reduction is mainly for dimension reduction and feature extraction. 

This study uses the Scikit-learn algorithm to select a path to infer the appropriate algorithm [12]. With a dataset sample size greater than 50, the data were classified and labelled. We realized that this study was suitable for classification problems. Accordingly, the classification algorithm model for supervised learning was used. We also used the regression and clustering algorithm in this algorithm [13].

We used 10 types of ML approaches—decision trees, random forest, Bernoulli naïve Bayes, multinomial naïve Bayes, Gaussian naïve Bayes, logistic regression, support vector machine, AdaBoost, quadratic discriminant analysis, and eXtreme gradient boosting (XGBoost) (Figure 1)—and selected the one with the highest accuracy for our complex datasets [10,11]. It is believed that, the more the learners, the greater the chance of providing optimal prediction under different circumstances. Gradient boosting is a technique that begins from shallow trees with a few leaves, reflecting low variance, but aggregates more trees with different parameters to reduce bias [14,15]. In our analysis, XGBoost demonstrated the highest accuracy, so we chose it as our ML technique (Figure 2). The introduction of XGBoost is provided in the Appendix A.

### 2.2. ML Model Prototype

We used the XGBoost technique to create a prototype to correlate patient and hospital characteristics to determine patient prognosis (Figure 3). For the use of emergency physicians, we created another recent-course algorithm to improve feasibility by removing variables such as hospitalization days, length ICU-staying days, and complications (Figure 4), which cannot be used soon after admission. In other words, the second model used variables that trauma surgeons or ED physicians are likely to have within a few days of admission.

## 3. Results

### 3.1. Description of Dataset

In the study period, 5968 patients were discharged from the ED, and the data of 5871 of them (57.9% of males) were used in this study. The patients’ ages ranged from 6 months to 103 years (mean: 48.8 ± 22.3 years), with 6.1% of patients being in the pediatric age group (<15 years) and 43.8% being in the geriatric age group (≥55 years); 37.7% of all patients had comorbidities. Furthermore, 51% of patients were transported to the ED by emergency medical services, 29.1% visited the ED themselves, and 19.9% were transferred from other hospitals. Among the 168 patients who died, 65.5% were transported to the ED by emergency medical service, 25.0% were transferred from other hospitals, and only 9.5% visited the ED themselves. The average level of triage was 2.4 ± 0.6, where we used triage classification from Taiwan Society of Emergency Medicine [16]. Moreover, 7.9% of all injured patients were classified as class I (most severe), 44.8% as class II (emergency), and 47.3% as class III (urgent). In all, 66.7% had a hospital stay of ≤7 days, 28.9% required hospitalization for 7 days to 1 month, and 4.4% required prolonged hospitalization over 30 days. The requirements for intensive care and surgery were 17.3% and 69.6%, respectively. Regarding patients with AIS scores ≥ 3, the extremities (24.4%) and head (21.4%) were the most commonly injured body parts, followed by the chest (8.5%), abdomen (3.2%), face (0.6%), and external (0.6%). The average ISS was 9.7 ± 8.8, new ISS was 11.7 ± 10.5, RTS was 7.656 ± 0.7414, and TRISS was 0.9543 ± 0.1267. Approximately 1 in 5 patients (20.7%) were classified as having major trauma (ISS > 15). Among the deceased, ISS < 15 was noted in 13.7% of patients, most of them being geriatric patients. Regarding prognoses, 69.4% patients recovered, 26.9% required chronic care, 0.9% were transferred to another hospital, and 2.8% died (Table 1).

### 3.2. Models

Two prediction models were tested in this study. The following are their high-ranking features.

### 3.3. Complete Model and High-Ranking Features

The complete model exhibited recalls of 86% for recovery, 30% for chronic care, and 67% for mortality (Table 2), and the accuracy of predicting complications was 80%. We noted the following 15 high-ranking features: trauma injury severity score(TRISS), AIS of the head, dispatch of emergency medical service, triage, new ISS, time from the injury site to the hospital, requirement for intensive care, length of hospital stay, Glasgow Coma Scale score in the ED, prehospital management by emergency medical technicians, RTS, prehospital consciousness, transportation from injury sites to the hospital, comorbidity, and deliberate or accidental injury.

### 3.4. Short-Term Model and High-Ranking Features

The short-term model displayed recalls of 89% for recovery, 25% for chronic care, and 41% for mortality. We noted the following 15 high-ranking features: trauma injury severity score, AIS of the head, dispatch of emergency medical service, time from injury sites to the hospital, new ISS, Glasgow Coma Scale score in the ED, triage, transportation from the injury site to the hospital, sex, prehospital management by emergency medical technicians, level of the transferred hospital, time from surgical decision to surgery, deliberate or accidental injury comorbidity, and age.

## 4. Discussion

The emergency medical system is well established in Taiwan. Most patients with severe trauma are transferred from the injury site to hospitals. In addition, as a regional trauma center, Chi-Mei Medical Center receives many patients with severe trauma transferred from neighboring hospitals. This explains why fewer than 10% of all deceased patients visited the hospital by themselves; most of them arrived by ambulance. 

In this study, we attempted to use as many clinical features as possible in this ML algorithm to predict the prognosis of patients with trauma. Traditional statistical methods cannot cope with large datasets that have complex, often nonlinear, characteristics between those variables and outcomes. By contrast, an ML algorithm can account for multiple variables at a time with a relatively higher predictive power [17,18]. We also acknowledged that emergency physicians and trauma surgeons may not have all the variables used in this ML algorithm in a short time. Therefore, we established a short-term model that included limited variables to resolve the problem. However, we believe this ML algorithm is clinically practical because almost every high-ranking feature in the complete and short-term model can be obtained within 24 h after arrival at the hospital.

If we plan to explore the use of this ML model to other hospitals, an external validation should be indicated. Nevertheless, patterns of traumatic injuries and management for patients with trauma differ between different areas or countries. A medical institution can apply a unique algorithm to predict patient outcomes and to improve the quality of care of patients with trauma.

We discovered that injury severity is the most decisive factor related to a patients’ prognosis. Accordingly, the high-ranking features of both models demonstrated that various trauma scores, prehospital setting, and patients’ conditions on arrival at the ED are crucial prognostic factors. Because the prognostic results of this ML algorithm included chronic care needs, injuries to the central nervous system, represented by the AIS of the head and Glasgow Coma Scale score in the ED, became a determinant of patient prognosis. Because the head is a commonly injured body part of the study cohort, injury to the head had a much higher importance than other body parts did [5,6,8,19].

We collected as many patients as possible in this study for better predictive performance with increased population size and variability, but the result was not as we expected. The first reason is that we did not categorize the different age populations well, for the impact of trauma would lead to different effects on diverse age groups, such as pediatric or geriatric patients [5]. The second probable reason is that most of our patients recovered rather than requiring chronic care or dying. Because citizens in Taiwan believe care is better in medical centers, many patients with mild wounds were sent to tertiary trauma centers, which made the ML method predict much more favorable outcomes. The third reason may contribute to high multicollinearity among the selected variables. Most trauma scores contain variables that have been included in building the predicted models, such as Glasgow Coma Scale, blood pressure, and respiratory rate. Finally, the imbalanced distribution of patients among recovery, chronic care, and death results in the low recall rate of both models. If we include only patients with severe trauma, as in most previous studies, this indicates that, for patients whose ISS > 15, the imbalanced distribution may be corrected. However, the adjustment will reduce the number of patients to 1233 and limit the development of this predicted model. Moreover, this model cannot be applied for all trauma patients, which does not meet the initial setting of this model.

We did not include unidentified influential parameters as variables, especially during hospitalization; therefore, predictive power did not correlate well with the variables [10]. A similar reason could explain that out-of-hospital cardiac arrest (OHCA) seemed to be an essential parameter, but it was not a high-ranking feature. We assumed that this finding was due to the small number of patients with OHCA status and the overlap between the OHCA-related and trauma score variables. Use of propensity score matching estimates and outcome-based regression models in ML may improve the correlation [20]. However, a larger dataset achieves better prediction because both data quality and quantity affect prediction power [10].

However, we were confronted with another issue, namely that mild disease constitutes most of the population, which might dilute our model, especially considering that we developed it to evaluate severe patients. Thus, future studies may select only patients who were transported to the ED by emergency medical services. This restriction can minimize insufficient resuscitation data between interfacility care [21] and improve exact records of the time of the accident, especially for patients who visit the ED by themselves or are transferred; results from the injury time to the arrival at the hospital would mostly be accurate, thereby minimizing the flaw from a high-ranking feature.

## 5. Limitations

First, our dataset was retrospective, and a prospectively collected database would be more complete and, thus, yield better performance. However, a large dataset can minimize the effect of missing data. Second, our study population was from a single urban trauma center, which may not be generalizable to other areas, particularly suburban or rural areas [6,21]. Moreover, unlike other similar artificial intelligence studies, we did not include laboratory variables, such as hemoglobin, glucose creatinine, and alanine aminotransferase [6], in order to achieve a rapidly useable model and enable quick clinical decision making; nevertheless, this may have lowered the predictive power of our model, especially that of the complete model. Third, we could discover trauma scores determining high-ranking features, but many variables were common among them; thus, interaction among variables and trauma scores should be reduced by setting a kernel or pretreating variables. Finally, new ML techniques rapidly emerge and are used in various fields outside medicine; thus, a better ML algorithm for this particular problem than the ones we chose might be available [22].

## 6. Conclusions

ML can assist health care professionals in clinical decision making because it can manage multiple variables simultaneously and find relationships between them [9]. We selected the XGBoost method to analyze the data of nearly 6000 cases within 2 years and found a good recall rate for patients with trauma who recovered (86% and 89% with the complete and short-term models, respectively) and those who experienced complications (80%). However, the recall was unsatisfactory for those requiring chronic care or those who died, likely owing to their small population in our cohort. Furthermore, TRISS, AIS of the head, dispatch of EMS, and new ISS were found to be high-ranking features in both models, with triage degree and time from injury site to the hospital specific to the complete and short-term models, respectively. In the future, we will attempt to categorize different age groups, expand our database, and diminish variables’ interaction within trauma scores to fit our model more appropriately to real-world scenarios.

## Figures and Tables

**Figure 1 medicina-58-01379-f001:**
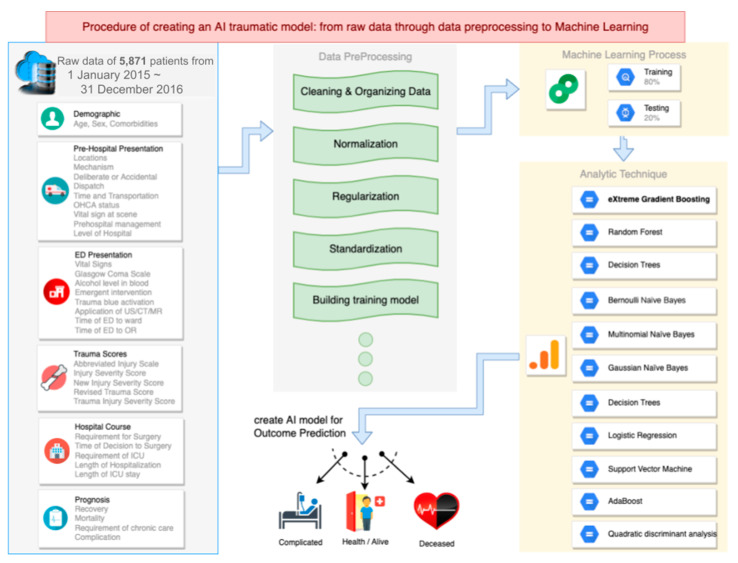
Flowchart from the database of beginning, through data optimization and training model to final clinical utilization.

**Figure 2 medicina-58-01379-f002:**
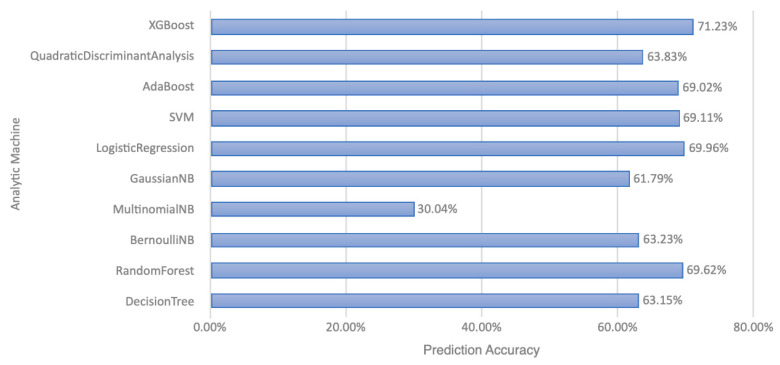
Comparing different analytic machine techniques for better predicting accuracy.

**Figure 3 medicina-58-01379-f003:**
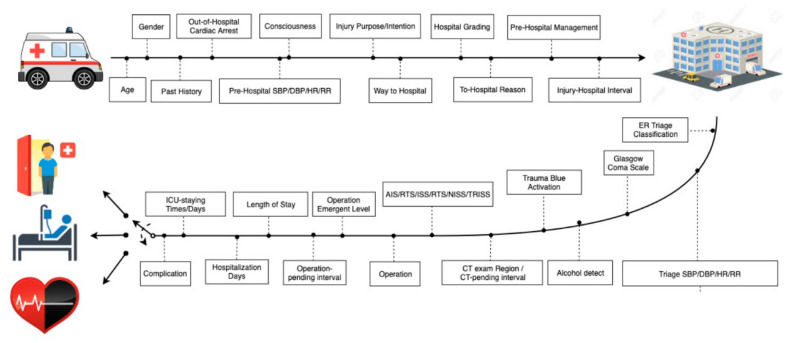
Algorithm of complete model: initial data collection from patient arriving through hospitalization to final prognosis prediction.

**Figure 4 medicina-58-01379-f004:**
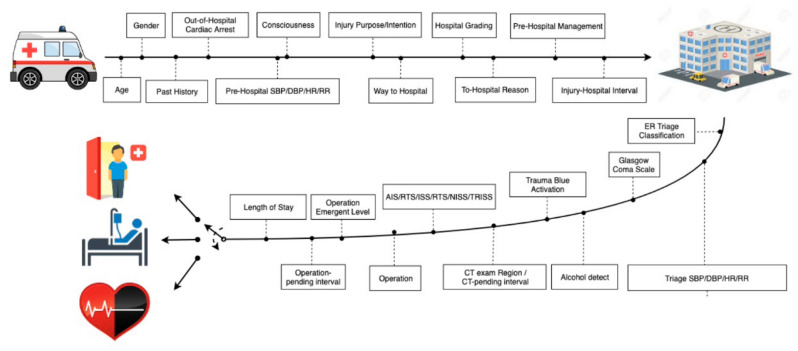
Algorithm of short-term model: initial data collection from patients arriving to prognosis prediction and skipping hospitalization.

**Table 1 medicina-58-01379-t001:** Clinical characteristics of study cohort.

**Age**	
Geriatric patients (>55 years)	43.8%
Non-geriatric adults	50.1%
Pediatric patients (<15 years)	6.1%
**Sex**	
Male	57.9%
Female	42.1%
**Transport type**	
By Emergency Medical Services	51%
By patient’s family or friends	29.1%
Transfer from other hospitals	19.9%
**Triage**	2.4 ± 0.6 (mean ± SD)
I	7.9%
II	44.8%
III	47.3%
**Hospital stays**	
<7 days	66.7%
7–30 days	28.9%
>30 days	4.4%
Requirement for intensive care	17.3%
Requirement for surgery	69.6%
**Abbreviated Injury Scale ≥ 3**	
extremities	24.4%
head	21.4%
chest	8.5%
abdomen	3.2%
face	0.6%
external	0.6%
Major trauma (ISS >15)	20.7%
**Trauma scores**	(mean ± SD)
Injury severity score (ISS)	9.7 ± 8.8
New injury severity score	11.7 ± 10.5
Revised trauma score	7.6560 ± 0.7414
Trauma injury severity score	0.9543 ± 0.1267

**Table 2 medicina-58-01379-t002:** Recall rate and high-ranking features of correlation of complete model and short-term model.

	Complete Model	Short-Term Model
Recovery	86%	89%
Chronic Care	30%	25%
Mortality	67%	41%
Complication	80%	
**High-ranking features**		
1	Trauma injury severity score	Trauma injury severity score
2	Abbreviated injury scale of head	Abbreviated injury scale of head
3	Dispatch of emergency medical service	Dispatch of emergency medical service
4	Triage	time from injury sites to the hospital
5	New injury severity score	New injury severity score
6	Time from injury sites to the hospital	Glasgow Coma Scale in ED
7	Requirement for intensive care	Triage
8	Length of hospital stay	transportation from injury sites to the hospital
9	Glasgow Coma Scale in ED	Gender
10	Prehospital managements by emergency medical technicians	Prehospital managements by emergency medical technicians
11	Revised trauma score	Level of transferred hospital
12	Prehospital consciousness	Time from surgical decision to surgery
13	Transportation from injury sites to the hospital	Deliberate or accidental injury
14	Comorbidity	Comorbidity
15	Deliberate or accidental injury	Age

## Data Availability

The source codes and data presented in this study are available on reasonable request to the cor-responding author (K.T.C.). The raw data are not publicly available because of a data protection policy for patient data and/or patent.

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
