# Peer review of "Prediction of Prognosis in Patients with Trauma by Using Machine Learning"

_medicina, 2022, doi:10.3390/medicina58101379_

Round 1

Reviewer 1 Report

I congrats to the authors for their effort.The severity of the trauma, as mentioned in the text, depends on many factors. Therefore, it is very difficult to find a prediction model. I would suggest performing further studies by dividing the sample by age and characteristics (e.g., the presence of comorbidities). However, I commend the attempt to use artificial intelligence in a field where it is still little known and could make a difference. Nice idea! 

Author Response

Responses to Reviewers’ comments

Open Review

(x) I would not like to sign my review report

( ) I would like to sign my review report

English language and style

( ) Extensive editing of English language and style required

( ) Moderate English changes required

(x) English language and style are fine/minor spell check required

( ) I don't feel qualified to judge about the English language and style

Yes   Can be improved    Must be improved  Not applicable

Does the introduction provide sufficient background and include all relevant references?

(x)    ( )     ( )     ( )

Are all the cited references relevant to the research?

(x)    ( )     ( )     ( )

Is the research design appropriate?

( )     (x)    ( )     ( )

Are the methods adequately described?

(x)    ( )     ( )     ( )

Are the results clearly presented?

( )     (x)    ( )     ( )

Are the conclusions supported by the results?

( )     (x)    ( )     ( )

Comments and Suggestions for Authors

I congrats to the authors for their effort. The severity of the trauma, as mentioned in the text, depends on many factors. Therefore, it is very difficult to find a prediction model. I would suggest performing further studies by dividing the sample by age and characteristics (e.g., the presence of comorbidities). However, I commend the attempt to use artificial intelligence in a field where it is still little known and could make a difference. Nice idea!

Response: Thank you for your comment. We will proceed to the next study according to your suggestions. Thank you.

Reviewer 2 Report

Dear authors,

the paper you wrote is interesting, mostly because of the high volume of enrolled patients, but I've some concerns.

1.Machine learning (ML) represents a new frontier among statistical models for predicting risks, and it should be more reliable than linear models, unfortunately in your paper the description of the model you applied is not immediately understandable, mostly by those who have no dimility with this knowledge. Please, try to explain more extensively and in a more understandable manner how your models works

2. Could you provide, maybe as supplemental materials, description of how XG Boost has been identified as the best model among 10?

3. Both models require  multiple variables to be used, and they should not of immediate usage. Moreover they should require an external validation to prove their goodness

4. How could you explain the low recall rates of both models in predicting mortality and the need of chronic care? Could it depend on high multicollinearity among the selected variables? Please try to improve this aspect in "Discussion"

5. Major trauma patients in your population represent only 20% of cases. Do you expect different recall rates in mortality, chronic care if model applied in patients with ISS > 15?

Author Response

Responses to Reviewers’ comments

Comment 1: Open Review

(x) I would not like to sign my review report

( ) I would like to sign my review report

English language and style

( ) Extensive editing of English language and style required

(x) Moderate English changes required

( ) English language and style are fine/minor spell check required

( ) I don't feel qualified to judge about the English language and style

Yes   Can be improved    Must be improved  Not applicable

Does the introduction provide sufficient background and include all relevant references?

( )     (x)    ( )     ( )

Are all the cited references relevant to the research?

( )     (x)    ( )     ( )

Is the research design appropriate?

( )     ( )     (x)    ( )

Are the methods adequately described?

( )     ( )     (x)    ( )

Are the results clearly presented?

( )     ( )     (x)    ( )

Are the conclusions supported by the results?

( )     ( )     ( )     ( )

Comments and Suggestions for Authors

Dear authors,

the paper you wrote is interesting, mostly because of the high volume of enrolled patients, but I've some concerns.

Machine learning (ML) represents a new frontier among statistical models for predicting risks, and it should be more reliable than linear models, unfortunately in your paper the description of the model you applied is not immediately understandable, mostly by those who have no dimility with this knowledge. Please, try to explain more extensively and in a more understandable manner how your models works.

Response: Thank you for your remarkable comment. We added 2 paragraphs in section Method, page 5, line 3-17 and 2 additional references (reference 12, 13) to describe how this model works.

Comment 2: Could you provide, maybe as supplemental materials, description of how XG Boost has been identified as the best model among 10?

Response: In response to your valuable comment. We added a Supplementary Appendix to describe how XGBoost has been identified as the best model among 10.

Comment 3: Both models require multiple variables to be used, and they should not of immediate usage. Moreover they should require an external validation to prove their goodness

Response: We explain the problems regarding multiple variables and external validation in section Discussion, page 9, line 13-23.

Comment 4: How could you explain the low recall rates of both models in predicting mortality and the need of chronic care? Could it depend on high multicollinearity among the selected variables? Please try to improve this aspect in "Discussion"

Response: Thank you for your remarkable comment. We added two paragraphs in section Discussion, page 10, line 18-23 for the low recall rate of both models.

Comment 5: Major trauma patients in your population represent only 20% of cases. Do you expect different recall rates in mortality, chronic care if model applied in patients with ISS > 15?

Response: In response to your valuable comment. We added a paragraph in section Discussion, page 10, line 23-24 and page 11, line 1-3 to discuss this issue.

All the changes of the manuscript were written in red letters.

Round 2

Reviewer 2 Report

Dear Authors, thank you very much for the improvement of the paper. Now, it is more understandable mostly in those technical parts describing the statistic model you used.